



# Quality assurance and quality control of atmospheric organosulfates measured using hydrophilic interaction liquid chromatography (HILIC).

Ping Liu[1,2,4], Xiang Ding[1,3*], Yu-Qing Zhang[1], Daniel J. Bryant[2*], Xin-Ming Wang[1,3]

[1]State Key Laboratory of Organic Geochemistry and Guangdong Provincial Key Laboratory of Environmental Protection and Resources Utilization, Guangzhou Institute of Geochemistry, Chinese Academy of Sciences, Guangzhou, 510640, China

[2]Wolfson Atmospheric Chemistry Laboratories, Department of Chemistry, University of York, York YO10 5DD, U.K

[3]Guangdong-Hong Kong-Macao Joint Laboratory for Environmental Pollution and Control, Guangzhou Institute of Geochemistry, Chinese Academy of Science, Guangzhou, 510640, China

[4]University of Chinese Academy of Sciences, Beijing, 100049, China

*Correspondence to*: Xiang Ding (xiangd@gig.ac.cn)

**Abstract.** As a crucial constituent of fine particulate matter ($PM_{2.5}$), secondary organic aerosols (SOA) influence public health, regional air quality, and global climate patterns. This paper highlights the use of Hydrophilic interaction liquid chromatography (HILIC) which effectively retains strongly polar analytes that might exhibit incomplete or no retention in reverse chromatography, resulting in superior separation efficiency.

A HILIC column was used to analyze 7 standards, environmental standards (1648a and 1649b), and samples collected in urban environments in the Pearl River Delta region of Guangzhou. That serve as valuable reference points for evaluating the organic composition of the atmospheric environment. The results indicate a high degree of accuracy in the analytical method, sodium octyl-$d_{17}$ sulfate serves as the internal standard, with a linear correlation coefficient of the 7 standards, boasting a linear correlation coefficient R ranging from 0.987-0.999 and a slope, k, of the linear equation from 0.9662-2.2927. The instrument detection limit (IDLs) is established at 0.0026-0.0300 μg mL$^{-1}$, while the method detection limit (MDLs) falls within the range of 0.0077-0.2300 ng m$^{-3}$, demonstrating the method's exceptional sensitivity.

Since isoprene sulphates are highly polar due to containing a hydrophilic bond to the hydroxyl group and a hydrophobic bond to the sulphate, and as such showed strong retention using this method. This technique employs Sodium ethyl sulfate and Sodium octyl sulfate standards for semi-quantitative compound analysis isoprene-derived OSs, the error in sample analysis (EA) ranged from 12.25-95.26%



and the two standards maintaining a consistent recovery rate between 116%-131% and 86%-127%. These
findings indicate a high level of precision when semi-quantifying compounds with similar structural
characteristics, affirming the analysis method's minimal relative error and underscoring its repeatability,
process stability, and the reliability of its results for isoprene OSs. To enhance the method's reliability
assessment, the study analyzed polar organic components of standard particulate matter samples (1648a
and 1649b), providing precise determinations of several isoprene OSs using this method. Methyltetrol
sulfate (m/z 215) is the highest concentration in the ambient samples, up to 67.33 ng m$^{-3}$ at daytime.
These results serve as valuable reference points for assessing the organic composition of the atmospheric
environment.
**1. Introduction**
Organosulfates (OSs) represent a category of organic compounds featuring the sulfate functional group
(R-OSO$_3$H), found ubiquitously in atmospheric aerosols, OSs contribute to 5-30% of the organic mass
fraction within particulate matter(Shakya and Peltier, 2013; Shakya and Peltier, 2015; Tolocka and
Turpin, 2012; Surratt et al., 2008; Lukacs et al., 2009). Their unique hydrophilic and hydrophobic
characteristics influence the hygroscopicity and cloud condensation nuclei (CCN) formation potential of
aerosol particles (Hansen et al., 2015), underscoring the need for a comprehensive investigation into their
chemical compositions and formation mechanisms in the atmosphere. OSs are formed from the oxidation
of anthropogenic precursors, such as benzene and toluene and biogenic volatile organic compounds
(VOCs) such as isoprene, monoterpenes (primarily α-pinene, β-pinene, and limonene), sesquiterpenes,
aromatics, aldehydes, and others, under a variety of oxidation and sulfuric acid conditions(Surratt et al.,
2008; Surratt et al., 2010). Isoprene is the most abundant precursor of global secondary organic aerosol
(SOA)(Bates and Jacob, 2019; Hodzic et al., 2016). The epoxide pathway plays a critical role in isoprene
SOA (iSOA) formation, in which isoprene epoxydiols (IEPOX) and/or hydroxymethyl-methyl-α-lactone
(HMML) can react with nucleophilic sulfate producing isoprene-derived organosulfates (iOSs)(Surratt
et al., 2010; Lin et al., 2013; He et al., 2018).
Previous research has employed reversed-phase liquid chromatography (RPLC) for the analysis of
aqueous atmospheric samples encompassing water-soluble and methanol-extractable aerosol constituents,
as well as fog water(Bryant et al., 2020; Bryant et al., 2021). This reversed-phase approach, utilizing a



non-polar stationary phase and a polar mobile phase, effectively retains higher-molecular weight OSs
derived from monoterpenes (e.g., $C_{10}H_{16}NSO_7^-$) (Gao et al., 2006; Surratt et al., 2007b) and aromatic
OSs (e.g., $C_7H_7SO_4^-$) (Kundu et al., 2010; Staudt et al., 2014). However, it is less efficient for the
separation of lower-molecular weight and highly polar OSs, which elute in less than 2.5 minutes and co-
elute with various other OSs, small organic acids, polyols, and inorganic sulfates (Stone et al., 2012).
The co-elution of so many analytes leads to matrix effects, reducing the analyte's signal(Bryant et al.,
2020; Bryant et al., 2021; Bryant et al., 2023b; Bryant et al., 2023a). The isoprene-derived OSs are
hydrophilic compounds owing to their hydroxyl functional groups, and the organosulfates are ionic polar
compounds. Hence, an alternative approach for the isoprene-derived OSs characterization that could
accomplish simultaneous analysis of polar and water-soluble components while avoiding the drawbacks
associated with current analytical methods would be highly desirable.
To address this challenge, a Hydrophilic interaction liquid chromatography (HILIC) featuring an
amide stationary phase has been utilized (Hettiyadura et al., 2015; Hettiyadura et al., 2017; Cui et al.,
2018). HILIC is purposefully designed to retain molecules with ionic and polar functional groups and
has demonstrated effectiveness in retaining carboxylic acid-containing OSs like glycolic acid sulfate and
lactic acid sulfate, which are among the most prevalent atmospheric OSs quantified to date (Olson et al.,
2011; Hettiyadura et al., 2015; Hettiyadura et al., 2017; Cui et al., 2018). Since these OSs compounds
are easily ionized in negative mode, they can be efficiently detected in ESI-MS negative ionization mode
(Romero and Oehme, 2005; Surratt et al., 2007a). In this experiment, a combination of HILIC
chromatographic separation and tandem mass spectrometry (MS/MS) was employed to separate and
detect highly polar OSs relevant to the atmosphere. A mixed standard of OSs facilitated the separation,
identification, and quantification of polar, ionic, and non-volatile OSs present in the atmosphere. The
HILIC separation was accomplished using a (BEH) amide column, and OSs were semi-quantified based
on the calibration curve derived from alternative standards through tandem quadrupole mass
spectrometry detection (TQD). This approach enabled the detection and quantification of OSs originating
from isoprene within the atmosphere of the Pearl River Delta.
Recent studies have identified hundreds of OSs in the ambient environment (Iinuma et al., 2007;
Surratt et al., 2008; Riva et al., 2016; Brueggemann et al., 2017; Le Breton et al., 2018; Hettiyadura et
al., 2019; Bruggemann et al., 2019). Yet, authentic standards for OSs remain scarce, with only a few



commercially available or synthesized in laboratories (Staudt et al., 2014; Hettiyadura et al., 2015; Huang
et al., 2018). The utilization of different surrogate standards results in considerable discrepancies in
quantifying OS concentrations (Zhang et al., 2022; He et al., 2018; Surratt et al., 2008), signifying the
persisting challenge of accurate quantification in OS studies. HILIC chromatography is a promising
analytical technique for the separation of organosulfates from one another and the complex aerosol matrix.
When coupled with authentic standard development and highly sensitive MS/MS detection, it offers an
improved method for quantifying and speciating atmospheric organosulfates. Enhanced measurements
of this compound class will contribute to a better understanding of SOA precursors and their formation
mechanisms.

## 2 Experimental sections

### 2.1 Field Sampling

Sampling was undertaken during October 2018 in Guangzhou (GZ), GZ is situated in the Pearl River
Delta (PRD) region of southern China which has large-scale land coverage of broadleaf evergreen trees
as well as high-temperature and strong solar radiation all year round.
Field sampling was conducted using a $PM_{2.5}$ sampler (Tisch Environmental Inc., Ohio, USA) equipped
with quartz filters (8 in.×10 in.) at a flow rate of $1.1 m^3\ min^{-1}$. Additionally, field blanks were collected
at a monthly interval. Blank filters were covered with aluminum foil, and baked at 500℃ for 24 h to
remove organic material, Pre - and post - sampling flow rates were measured with a calibrated rotameter.
All filters were handled using clean techniques, which included storage of filters in plastic petri dishes
lined with pre-cleaned aluminum foil and manipulation with pre-cleaned stainless steel forceps. Post-
sampling, filters were stored frozen in the dark. One field blank was collected for every five samples.
and stored in a container with silica gel. After sampling, the filter samples were stored at -20℃.

### 2.2 PM sample extraction and preparation

Following the procedure outlined by Hettiyadura et al. (Hettiyadura et al., 2015), an 82 mm diameter
circular section was excised from the quartz membrane using a cutter. This section was subsequently cut
into small pieces with forceps that had been cleaned with ACN. The samples were then carefully placed
into a 100 mL clean beaker. To this, 200 μL of a solution with ACN and ultra-pure water (95:5, by



volume) containing sodium octyl-d$_{17}$ sulfate at a concentration of 5.3 μg mL$^{-1}$ was introduced as an
internal standard. Subsequently, 15 mL of acetonitrile (ACN) of chromatographic purity and ultrapure
water (95:5, by volume) were added in three separate increments. with the beaker was covered with
aluminum foil to prevent the organic solvent from evaporating, and extracted by ultra-sonication
extraction in an ice water bath for 20 minutes. The resulting solution was then filtered through a
polypropylene membrane syringe filter (0.45 μm; 25 pp, Sigma-Aldrich) and the process was repeated
three times to consolidate the solution. The solution was then concentrated to an approximate volume of
5 mL using a rotary evaporator, these were transferred to 1.5 mL vials and the solvent was blown to
dryness using a micro-scale nitrogen evaporation system at 35°C under a high-purity nitrogen stream,
Extracts were then re-constituted with ACN and ultra-pure water (95:5, by volume) to a final volume of
300 μL. The solution was thoroughly mixed and then stored in a freezer at -20°C for subsequent analysis.
**2.3 Instrumentation and Reagents**
OS sample analysis was performed using ultra-performance liquid chromatography electrospray triple
quadrupole tandem mass spectrometry (UPLC/ESI-MS/MS, Agilent 6400, USA) with a BEH amide
column (2.1 mm×100 mm, 1.7 μm; ACQUITYUPLC, Waters) in full-scan mode. The column
temperature was held at 35°C and the mobile phase flow rate was 0.5 mL min$^{-1}$. The injection volume of
samples and standards is 5 μL. Mobile phase A (organic phase) is 95:5 ACN: water (by volume) buffered
with ammonium acetate buffer (10 mm, pH 9) and mobile phase B (aqueous phase) is 100 % water,
ammonium acetate buffer (10 mm, pH 9). Use the MassHunter software (version B.02) to acquire and
process all data.

136       Purchased standards: Sodium methane sulfonate (>98.0%, Aladdin), sodium methyl sulfate (98%,

Sigma-Aldrich), Sodium ethyl sulfate (> 98%, Sigma-Aldrich), Sodium octyl sulfate (99%, Alfa Aesar),
Sodium dodecyl sulfate (99.0%, Sigma-Aldrich), Sodium hexadecyl sulfate (99%, Alfa Aesar), Sodium
octadecyl sulfate (99%, Alfa Aesar), Sodium octyl-d$_{17}$ sulfate (99.1%,   CDN), chromatographic pure
acetonitrile (ACN), (99.9%, CNW), ammonium acetate (99.0%, CNW), ammonia (Ammonia, 20%-22%,
CNW).



### 2.4 Separation and detection of organosulfates

### 2.4.1 Separation

For the separation process, ultra-high performance liquid chromatography electrospray triple quadrupole tandem mass spectrometry (UPLC/ESI–MS/MS) was employed, and the separation was optimized using a gradient elution method. Mobile phase A remained at 100% from 0 to 2 minutes, after which it decreased to 85% from 2 to 4 minutes and remained constant at 85% until the 11th minute. To re-equilibrate the column before the next injection, mobile phase A was reinstated to 100% between the 11th and 11.5th minute, and this composition was maintained until the 20th minute. The cleaning needle solvent employed a mixture of acetonitrile and ultrapure water (in a volume ratio of 80:20).

### 2.4.2 Detection

In the negative ion mode, the identification of organosulfates was achieved via TQD-MS, specifically utilizing an ACQUITY system by Waters as the mass spectrometer. The detector operated in Full Scan mode, with the first quadrupole selecting deprotonated molecules, the second quadrupole identifying fragments, and the third quadrupole analyzing product ions.

### 2.4.3 Optimization of experimental conditions

The choice of the fragmentation voltage directly impacts the instrument's ability to target specific compounds, while the collision energy plays a crucial role in determining the extent of fragmentation and the response of secondary fragment ions. To illustrate, when analyzing the most common compounds in the sample, and without connecting the chromatographic separation column, a 5 µL aliquot of the environmental sample was injected every 0.7 minutes. In this production scanning mode, the target ions generated after ionization in the ion source were detected. The first fragmentation voltage was set at 80 V, and with each subsequent scan, the voltage was incrementally increased by 5 V until it reached 180 V. The analysis revealed that the optimal response was achieved at 135 V. Consequently, 135 V was selected as the optimal fragmentation voltage for quantitative analysis of the actual samples.

For compounds with intricate chemical structures, further analysis was carried out using tandem mass spectrometry. Similarly, an energy level of 8 eV was employed in the collision cell during the OS daughter ion scanning. Table 1 displays the optimal fragmentation voltage and collision energy for different standards. The determination of other optimal conditions for the ESI source followed a similar





methodology, as presented in Table 2.
**Table 1 Optimal fragmentation voltage and collision energy of different standards.**

| Standards | Mass [M-H]⁻ | Fragmentation voltage(V) | Collision energy (eV) |
|---|---|---|---|
| Sodium methane sulfonate | 118.09 | 120-140 | 8 |
| Sodium methyl sulfate | 134.08 | 130-150 | 8-10 |
| Sodium ethyl sulfate | 148.11 | 130-150 | 8-10 |
| Sodium octyl sulfate | 232.27 | 120 | 8 |
| Sodium dodecyl sulfate | 288.38 | 130-150 | 8-10 |
| Sodium hexadecyl sulfate | 344.49 | 130-150 | 8-10 |
| Sodium octadecyl sulfate | 372.54 | 140 | 8-10 |
| Sodium octyl-$d_{17}$ sulfate | 232.27 | 120-140 | 8 |

**Table 2 Other ESI conditions of mass spectrometry.**

| Other ESI sources | Conditions |
|---|---|
| Source Gas Temp | 150℃ |
| Source Gas Flow | 1.7 L/min |
| Nebulizer | 45 psi |
| Sheath Gas Temp | 400℃ |
| Sheath Gas Flow | 12 L/min |
| Capillary | 2700 V |
| Nozzle Voltage | 500 V |
| Chamber Current | 0.18 µA |

**3 Results and discussion**
**3.1 Linearity of the standard**
In this experiment, a series of internal standards were employed, including the sodium octyl-$d_{17}$ sulfate
standard solution (200 µL; 5.3 µg mL⁻¹). The linear range of each standard solution was determined based
on its concentration ratio and peak area ratio. The standard curves of various compounds were evaluated
for their correlation coefficients, resulting in values ranging from 0.987 to 0.999. Notably, the standard
curve for octyl sodium sulfate exhibited a correlation coefficient (R) of 0.999, with a slope (k) of 0.9662,



indicating that the semi-quantification of structurally similar compounds using sodium octyl sulfate as
the standard was more precise when sodium octyl-$d_{17}$ sulfate was used as the internal standard.
**Table 3 The Linear and RSD of standards. R: Correlation coefficient, P: Pearson significance test.**

| Standards | Mass [M-H]- | Formula | $t$R(min) | k (slope) | R | P |
|---|---|---|---|---|---|---|
| Sodium methane sulfate | 118 | $CH_3SO_3^-$ | 4.815 | 2.2927 | 0.989 | 0.001 |
| Sodium methyl sulfate | 134.08 | $CH_3OSO_3^-$ | 1.064 | 1.4992 | 0.997 | 0.000 |
| Sodium ethyl sulfate | 148.11 | $C_2H_5OSO_3^-$ | 0.951 | 1.1849 | 0.987 | 0.002 |
| Sodium octyl sulfate | 232.3 | $CH_3(CH_2)_7OSO_3^-$ | 0.628 | 0.9662 | 0.999 | 0.000 |
| Sodium dodecyl sulfate | 288.38 | $CH_3(CH_2)_{11}OSO_3^-$ | 0.584 | 1.4836 | 0.995 | 0.000 |
| Sodium hexadecyl sulfate | 344.49 | $CH_3(CH_2)_{15}OSO_3^-$ | 0.567 | 1.8816 | 0.996 | 0.000 |
| Sodium octadecyl sulfate | 372.55 | $CH_3(CH_2)_{17}OSO_3^-$ | 0.558 | 1.3356 | 0.998 | 0.000 |

**3.2 UPLC/ESI–MS/MS instrument detection limits and method detection limits**
To ensure the effectiveness of this method in monitoring the target compounds in field environmental
samples, this study used 7 commercially available OS standards, and a regression analysis was conducted
using peak area as the x-axis and standard solution concentration as the y-axis. The resulting slope was
denoted as 'k.' The standard deviation (SD) was computed by repeatedly injecting the standard sample
with the lowest concentration six times in succession. The instrumental detection limits (IDLs) were
established at the 95% confidence interval, calculated as 3 times SD divided by 'k.' In this experiment,
with a sample sampling volume of 270 $m^3$ and considering the entire laboratory analysis process, the
method detection limits (MDLs) for these compounds were determined, as depicted in Table 4.
Of the various standard samples analyzed, the compound with the highest method detection limit was
sodium mesylate, which measured at 0.23 ng $m^{-3}$. This finding underscores the method's remarkable
sensitivity in detecting organosulfates in environmental aerosols, thereby affirming its effective detection
capability.
**Table 4 The IDLs: Instrumental detection limits (µg mL$^{-1}$) and MDLs: Method detection limits (ng m$^{-3}$) of**
**different standards. M: Sample concentration (µg mL$^{-1}$), total six times. SD: Standard deviation.**

| Standards | M1 | M2 | M3 | M4 | M5 | M6 | SD | IDLs (µg mL$^{-1}$) | MDLs (ng m$^{-3}$) |
|---|---|---|---|---|---|---|---|---|---|





| | | | | | | | | | |
|---|---|---|---|---|---|---|---|---|---|
| Sodium methane sulfonate | 0.042 | 0.042 | 0.029 | 0.047 | 0.052 | / | 0.009 | 0.0300 | 0.2300 |
| Sodium methyl sulfate | 0.024 | 0.023 | 0.022 | 0.020 | 0.024 | 0.022 | 0.0014 | 0.0041 | 0.0123 |
| Sodium ethyl sulfate | 0.016 | 0.015 | 0.014 | 0.014 | 0.015 | 0.018 | 0.0015 | 0.0044 | 0.0130 |
| Sodium octyl sulfate | 0.007 | 0.006 | 0.005 | 0.005 | 0.005 | 0.005 | 0.0009 | 0.0026 | 0.0077 |
| Sodium dodecyl sulfate | 0.025 | 0.022 | 0.022 | 0.024 | 0.026 | 0.026 | 0.0020 | 0.0059 | 0.0174 |
| Sodium hexadecyl sulfate | 0.047 | 0.043 | 0.041 | 0.043 | 0.045 | 0.048 | 0.0026 | 0.0079 | 0.0234 |
| Sodium octadecyl sulfate | 0.023 | 0.020 | 0.018 | 0.020 | 0.021 | 0.020 | 0.0017 | 0.0050 | 0.0149 |

Note: MDL is calculated for the sample form Guangzhou.
**3.3 Parallelism and recovery of experiments**
In this experiment, a matrix spike experiment was conducted. Approximately 200 μL of a mixed
solution, containing all the standards at a concentration of around 5 μg mL$^{-1}$, was injected onto a 47 mm
blank quartz membrane. This procedure was repeated in parallel five times, and a sample without the
mixed solution served as a laboratory blank, adding up to a total of six sample groups for pretreatment
analysis. The total quantity of each substance in the treated sample and the content of each substance in
the untreated sample were computed, thereby enabling the calculation of the recovery rate for each
compound. As demonstrated in Table 5, the recovery rates for various compounds fell within the range
of 54% - 146%. These high recovery rates indicate minimal loss of the target compounds during the
analysis, which is favorable for accurate detection.
Moreover, it is noteworthy that the relative deviation for these standards did not surpass 15%,
underscoring the small relative error and highlighting the experiment's reproducibility. The Relative
standard deviations (RSDs) of the small molecule were all less than 5.87%, but the RSDs for long-chain
alkane OSs are all higher than 10%, this indicating that this experiment is favourable for the detection of
OSs of isoprene. The stability of the analysis process ensures that the results obtained are reliable.
**Table 5 The recovery and RSD (Relative standard deviation) of standards. M: Sample recovery (%).**

| Standards | M1(%) | M2(%) | M3(%) | M4(%) | M5(%) | Recovery | RSD (%) |
|---|---|---|---|---|---|---|---|
| Sodium methane sulfonate | 63.31 | 62.78 | 54.81 | 60.23 | 62.93 | 54%-64% | 5.87 |
| Sodium methyl sulfate | 61.44 | 64.64 | 60.32 | 60.55 | 60.20 | 60%-65% | 3.03 |
| Sodium ethyl sulfate | 127.84 | 130.89 | 116.43 | 122.78 | 126.12 | 116%-131% | 4.43 |



| | | | | | | |
|---|---|---|---|---|---|---|
| Sodium octyl sulfate | 126.85 | 101.24 | 105.97 | 108.68 | 86.41 | 86%-127% | 13.76 |
| Sodium dodecyl sulfate | 145.16 | 131.76 | 111.62 | 113.06 | 100.26 | 100%-146% | 14.85 |
| Sodium hexadecyl sulfate | 121.27 | 119.10 | 114.12 | 114.79 | 87.90 | 87%-122% | 12.10 |
| Sodium octadecyl sulfate | 117.23 | 95.04 | 108.21 | 86.71 | 84.45 | 85%-118% | 14.33 |

### 3.4 Empirical approach to estimate error in sample analysis

Stone et al. (Stone et al., 2012) developed an empirical approach to estimate the error resulting from surrogate quantification ($E_Q$) based on a homologous series of atmospherically relevant compounds. They estimated the relative error introduced by each carbon atom ($E_n$), oxygenated functional group ($E_f$), and alkenes ($E_d$) to be 15%, 10%, and 60%, respectively. The errors introduced by surrogate quantification are considered additive and are calculated as follows. Furthermore, the error in sample analysis ($E_A$) can be estimated through the error propagation of field blank ($E_{FB}$), spike recovery ($E_R$), relative differences ($E_D$), and the surrogate quantification ($E_Q$) calculated following Eq. (1). The error in sample analysis ($E_A$) calculated following Eq. (2):

$$\%E_Q = \%E_n \Delta n + \%E_f \Delta f + \%E_d \Delta d \tag{1}$$

$$\%E_A = \sqrt{(\%E_{FB})^2 + (\%E_R)^2 + (\%E_D)^2 + (\%E_Q)^2} \ldots \tag{2}$$

Where $\Delta n$ represents the difference in the number of carbon atoms between a surrogate and an analyte, $\Delta f$ is the difference in oxygen-containing functional groups between a surrogate and an analyte, and $\Delta d$ is the difference in alkene functionality between a surrogate and an analyte. As shown in Table 6, the $E_Q$ ranged from 10% to 95% for the OSs when using sodium ethyl sulfate and sodium octyl sulfate as the surrogates. The $E_Q$ values were compared to the previous surrogate with camphorsulfonic acid, there is 215% and 230% reduced to 75% and 60% for 2-MTOOS (m/z 215) and 2-MGAOS (m/z 199), respectively(Zhang et al., 2022). And $E_A$ ranged from 12.25-95.26% for these iOS products. For 2-MTOOS (m/z 215) and 2-MGAOS (m/z 199), $E_A$ are 73.33% and 60.42%, respectively (see Table 6).

**Table 6 Uncertainty associated with sample analysis.**

| Compounds [M-H]- | Formula | Surrogates | Surrogate formula | EQ(%) | EA(%) |
|---|---|---|---|---|---|
| 139 | $C_2H_3O_5S$ | Sodium ethyl sulfate | $C_2H_5O_4S$ | 10 | 12.25 |
| 153 | $C_3H_5O_5S$ | Sodium ethyl sulfate | $C_2H_5O_4S$ | 25 | 25.98 |
| 155 | $C_2H_3O_6S$ | Sodium ethyl sulfate | $C_2H_5O_4S$ | 20 | 21.21 |





| 167 | $C_4H_7O_5S$ | Sodium ethyl sulfate | $C_2H_5O_4S$ | 40 | 40.62 |
|---|---|---|---|---|---|
| 169 | $C_3H_5O_6S$ | Sodium ethyl sulfate | $C_2H_5O_4S$ | 35 | 35.71 |
| 183 | $C_4H_7O_6S$ | Sodium ethyl sulfate | $C_2H_5O_4S$ | 50 | 50.50 |
| 199 | $C_4H_7O_7S$ | Sodium octyl sulfate | $C_8H_{17}O_4S$ | 60 | 60.42 |
| 215 | $C_5H_{11}O_7S$ | Sodium octyl sulfate | $C_8H_{17}O_4S$ | 75 | 75.33 |
| 237 | $C_7H_9O_7S$ | Sodium octyl sulfate | $C_8H_{17}O_4S$ | 45 | 45.55 |
| 260 | $C_5H_{10}O_9NS$ | Sodium octyl sulfate | $C_8H_{17}O_4S$ | 95 | 95.26 |

### 3.5 MS$^2$ of isoprene organosulfates

In this experiment, the semi-quantitative determination of isoprene organosulfate was carried out using sodium octyl-d$_{17}$ sulfate as the internal standard, sodium ethyl sulfate and sodium octyl sulfate as the standards. Semi-quantitative analytical methods were employed to monitor the characteristic product ions of organosulfates (Stone et al., 2009), namely $HSO_4$ (m/z 97) and $SO_4$ (m/z 96). Tandem mass spectrometry (MS$^2$) was utilized as a means of identifying organosulfates and performing semi-quantitative analysis when actual standards were not available.

Given the wide array of polar compounds present in field samples and the substantial variations between samples, the final qualitative and quantitative analysis was carried out in full-scan mode. This approach ensured the most comprehensive component analysis results. By evaluating the relative signal intensity using HILIC-triple quadrupole mass spectrometry (TQD), it was possible to identify certain organosulfates.





**Figure 1: MS$^2$ diagram of isoprene-derived organosulfates.**

Note: Only one MS$^2$ is listed for reference





**3.6 Measurement of environmental standards**
The relatively pristine nature of the standard mixture solution stands in stark contrast to the actual field
ambient atmospheric aerosol samples, which are characterized by complex matrices that can significantly
influence the analytical results. To comprehensively assess the reliability of this analytical method, we
acquired standard particulate matter samples (NIST 1648a and 1649b). We proceeded to analyze the
organic components within these samples and determine the content of environmental standard particle
samples using the same method. The results, as presented in Tables 7 and 8, among them, the retention
time for isoprene organosulfates is all greater than the deadtime of the column, indicating that the method
provides good retention and separation for highly polar isoprene organosulfates, and reveal that the
relative deviation in the analysis of all compounds does not exceed 26.75%. This level of deviation falls
within the acceptable range for the analysis of organic compounds, affirming the method's suitability for
field sample analysis. These results serve as valuable reference points for assessing the organic
composition of the atmospheric environment.
**Table 7 The content and RSD of compounds in 1648a. M: Sample concentration (ng m$^{-3}$).**

| Compounds [M-H]$^-$ | M1 | M2 | M3 | M4 | M5 | Average | $t$R(min) | RSD |
|---|---|---|---|---|---|---|---|---|
| 139 ($C_2H_3SO_5^-$) | 15.02 | 17.82 | 14.73 | 13.01 | 13.97 | 14.91 | 0.83,1.58 | 12.11% |
| 153 ($C_3H_5SO_5^-$) | 26.60 | 29.11 | 24.72 | 23.65 | 24.77 | 25.77 | 0.79,0.82 | 8.33% |
| 155 ($C_2H_3SO_6^-$) | 1.83 | 1.94 | 1.76 | 1.78 | 1.42 | 1.75 | 0.74,1.47.1.79 | 11.08% |
| 167 ($C_4H_7SO_5^-$) | 17.27 | 15.76 | 14.60 | 14.28 | 15.48 | 15.48 | 0.69 | 7.57% |
| 169 ($C_3H_5SO_6^-$) | 1.58 | 1.90 | 1.57 | 1.27 | 1.53 | 1.57 | 1.46 | 14.29% |
| 183 ($C_4H_7SO_6^-$) | 9.30 | 10.05 | 8.31 | 7.97 | 8.69 | 8.86 | 0.86,1.00 | 9.31% |
| 199 ($C_4H_7SO_7^-$) | 5.62 | 6.71 | 6.18 | 5.49 | 5.77 | 5.95 | 10.22 | 8.33% |
| 215 ($C_5H_{11}SO_7^-$) | 70.03 | 84.46 | 81.43 | 68.00 | 79.89 | 76.76 | 1.83,2.34,4.25,5.24, 6.07,6.54 | 9.50% |
| 237 ($C_7H_9SO_7^-$) | 7.02 | 8.51 | 8.20 | 7.49 | 7.55 | 7.55 | 0.71 | 7.65% |
| 260 ($C_5H_{10}NSO_9^-$) | 7.95 | 10.98 | 6.06 | 6.00 | 7.18 | 7.63 | 0.65,1.02 | 26.75% |

**Table 8 The content and RSD of compounds in 1649b. M: Sample concentration (ng m$^{-3}$).**

| Compound [M-H]$^-$ | M1 | M2 | M3 | M4 | M5 | Average | $t$R(min) | RSD |
|---|---|---|---|---|---|---|---|---|





| | | | | | | | | |
|---|---|---|---|---|---|---|---|---|
| 139 ($C_2H_3SO_5^-$) | 22.53 | 26.21 | 24.22 | 25.04 | 22.41 | 24.08 | 0.83,1.58 | 6.78% |
| 153 ($C_3H_5SO_5^-$) | 37.65 | 36.64 | 39.90 | 39.77 | 35.13 | 37.82 | 0.79,0.82 | 5.42% |
| 155 ($C_2H_3SO_6^-$) | 2.24 | 2.08 | 2.24 | 2.28 | 1.88 | 2.15 | 0.74,1.47.1.79 | 7.79% |
| 167 ($C_4H_7SO_5^-$) | 22.24 | 23.13 | 23.82 | 23.55 | 20.63 | 22.67 | 0.69 | 5.69% |
| 169 ($C_3H_5SO_6^-$) | 1.99 | 2.42 | 2.73 | 2.42 | 2.34 | 2.38 | 1.46 | 10.99% |
| 183 ($C_4H_7SO_6^-$) | 7.22 | 8.78 | 8.12 | 8.27 | 7.79 | 8.04 | 0.86,1.00 | 7.24% |
| 199 ($C_4H_7SO_7^-$) | 0.04 | 8.11 | 0.04 | 7.16 | 6.67 | 4.40 | 10.22 | 91.12% |
| 215 ($C_5H_{11}SO_7^-$) | 98.60 | 131.24 | 114.14 | 115.53 | 106.45 | 113.19 | 1.83,2.34,4.25,5.24, 6.07,6.54 | 10.73% |
| 237 ($C_7H_9SO_7^-$) | 9.14 | 11.72 | 9.23 | 10.75 | 9.86 | 10.14 | 0.71 | 10.78% |
| 260 ($C_5H_{10}NSO_9^-$) | 3.06 | 3.36 | 3.75 | 3.25 | 3.13 | 3.31 | 0.65,1.02 | 8.21% |

**3.7 Isoprene-derived organosulfates in ambient PM samples**

Concentrations of isoprene-derived organosulfates quantified in ambient $PM_{2.5}$ from Guangzhou in October 2018 daytime and nighttime, are provided in Table 9. Methyltetrol sulfate (m/z 215) is the most prevalent OS known to date(Surratt et al., 2008; Hettiyadura et al., 2015). It is formed through a nucleophilic addition reaction involving an IEPOX ring, catalyzed by sulfuric acid (Surratt, Chan et al. 2010). $C_5H_{11}O_7S^-$ (m/z 215) exhibited peak retention times of 1.83, 2.34, 4.25, 5.24.6.07 and 6.54 minutes and was the most abundant organosulfate measured. On 7th October during the daytime and 7th-8th October during the nighttime, its concentrations were 67.33 ng m$^{-3}$ and 57.94 ng m$^{-3}$, respectively.

The OSs with formular m/z 260 is a nitroxic OS resulting from the photooxidation of isoprene under high NOx conditions (Gomez-Gonzalez et al., 2008; Surratt et al., 2008). In the course of this experiment, two isomers with an m/z 260 were discovered, with Hettiyadura and colleagues identifying two such isomers in 2019 (Hettiyadura et al., 2019), and Centreville identifying four isomers with m/z 260(Surratt et al., 2008). And an m/z 260 exhibits a moderate correlation with methyltetrol sulfate, hinting at isoprene as a likely precursor (Hettiyadura et al., 2019). In this experiment, the concentration of m/z 260 was significantly higher at night than during the day, were 17.55 ng m$^{-3}$ and 10.21 ng m$^{-3}$, respectively. Further subsequent experiments could explore the reasons for this diurnal difference in terms of the mechanism of formation of m/z 260.

Organosulfates with the formulas $C_4H_7O_7S^-$ (m/z 199, calculated mass: 198.9912) is an oxidation product of isoprene under high NOx conditions. In this method, the retention time for the peak is 10.22





minutes, and the concentration of m/z 199 was significantly higher at night than during the day, were
18.13 ng m$^{-3}$ and 12.51 ng m$^{-3}$, respectively, suggesting that nighttime chemistry is more conducive to
the formation of m/z 199.
In summary, these findings strongly suggest that isoprene serves as the primary and most abundant
precursor to OSs. Hettiyadura et al. (Hettiyadura et al., 2019) demonstrated that during the Atlanta
summer, over half of the organic aerosol compounds derived from isoprene are composed of OSs, with
methyltetrol sulfate being the predominant constituent, Subsequent experiments can further explore the
different formation mechanisms of these isoprene-derived organosulfates and the reasons for the
variations in different isomers.
**Table 9 Ambient concentrations of isoprene-derived organosulfates measured in PM$_{2.5}$ at Guangzhou, from**
**06:00-18:00 on 7/10/2018 (daytime) and 18:00-06:00 on 7/10/2018-8/10/2018 (nighttime).**

| m/z | Formula | Mass | tR (min) | Time | Concentration(ng m$^{-3}$) |
|---|---|---|---|---|---|
| 139 | $C_2H_3SO_5^-$ | 138.9701 | 0.83,1.58 | Daytime | 7.70 |
| | | | | Nighttime | 9.16 |
| 153 | $C_3H_5SO_5^-$ | 152.9858 | 0.79,0.82 | Daytime | 20.88 |
| | | | | Nighttime | 34.92 |
| 155 | $C_2H_3SO_6^-$ | 154.9650 | 0.74,1.47.1.79 | Daytime | 13.81 |
| | | | | Nighttime | 18.68 |
| 167 | $C_4H_7SO_5^-$ | 167 | 0.69 | Daytime | 4.82 |
| | | | | Nighttime | 7.66 |
| 169 | $C_3H_5SO_6^-$ | 168.9807 | 1.46 | Daytime | 11.02 |
| | | | | Nighttime | 11.75 |
| 183 | $C_4H_7SO_6^-$ | 182.9963 | 0.86,1.00 | Daytime | 8.80 |
| | | | | Nighttime | 8.69 |
| 199 | $C_4H_7SO_7^-$ | 198.9912 | 10.22 | Daytime | 12.51 |
| | | | | Nighttime | 18.13 |
| 215 | $C_5H_{11}SO_7^-$ | 215.0225 | 1.83,2.34,4.25,5.24, 6.07,6.54 | Daytime | 67.33 |
| | | | | Nighttime | 57.94 |
| 237 | $C_7H_9SO_7^-$ | 237 | 0.71 | Daytime | 11.00 |





| | | | | | |
|---|---|---|---|---|---|
| | | | | Nighttime | 15.37 |
| 260 | $C_5H_{10}NSO_9^-$ | 260.0076 | 0.65,1.02 | Daytime | 10.21 |
| | | | | Nighttime | 17.55 |

**4 Conclusion**

Organosulfates (OSs) are a vital component of Secondary Organic Aerosols (SOA). Previously, their measurement using reverse phase liquid chromatography presented challenges due to a lack of retention and subsequent co-elution with other organic sulfates, small organic acids, polyols, and inorganic ions, resulting in poor separation and matrix effects. In this experiment, we employed Hydrophilic Interaction Liquid Chromatography (HILIC) to analyze organosulfates in the atmospheric environment. HILIC effectively resolved this issue by delaying the elution time of molecules with ionic and polar functional groups, particularly OSs containing carboxyl groups. HILIC retained strongly polar samples that had incomplete or no retention in C18 reverse chromatography, offering a solution to the co-elution problem of organosulfates with other small compounds in C18 reverse columns, resulting in a robust separation.

During this experiment, we conducted isoprene-derived organosulfates in the atmospheric environment of the Pearl River Delta using HILIC. And our analytical method possessed high sensitivity, enabling effective detection of organosulfates in environmental aerosols. Each standard exhibited a relative deviation controlled within 15%, indicating minimal relative errors, high experimental reproducibility, stable analysis procedures, and reliable results. We also simultaneously analyzed two environmental reference standards (NIST 1648a and 1649b), providing some reference for the quantification of atmospheric organosulfates.

Nonetheless, research on OSs commenced relatively late, and due to their wide diversity and demanding laboratory synthesis conditions, only a limited number of commercial reference materials are available for quantitative OSs analysis. Consequently, the lack of actual standards led us to employ semi-quantitative analysis methods in this experiment, introducing some uncertainty in quantification. Future work should focus on enhancing the quantitative methods for OSs, utilizing actual standards for one-to-one compound quantification, and refining the measurement techniques for organosulfates. These efforts will contribute to a deeper understanding of SOA precursors, formation mechanisms, and the contribution of OSs to atmospheric aerosols, ultimately guiding research in the field of air pollution prevention and



control.
*Acknowledgements.* This research was supported by the Foundation for Innovative Research Groups
of the National Natural Science Foundation of China (42321003), National Natural Science Foundation
of China (42177090), and we thank for the financial support from the China Scholarship Council (CSC).

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
