# Peer review of "Quality assurance and quality control of atmospheric"

_Atmospheric Measurement Techniques, 2023_

## Referee Comment (RC1)

Authors developed a new method about the measurement of major organosulfates in the atmosphere (e.g., $C_4H_7SO_7^-$ and $C_5H_{11}SO_7^-$) that can separate them from other compounds in the column. Their results and findings provide a valuable insight to the folks who are interested in studying the chemical behaviors of organosulfates in the atmosphere. However, there are still some issues should be noted and addressed before the paper can be published on the AMT journal.

1. Authors pointed out that the previous method has the co-elution problem that affecting the quantification of OSs, especially for lower-molecular weight and highly polar OSs. Thus, they employed the method of HILIC using amide stationary phase to measure OSs, finding that this method can successfully separate some isoprene-derived OSs (i.e., $C_4H_7SO_7^-$ and $C_5H_{11}SO_7^-$) from other atmospheric OA components. However, as shown in Tables 7 and 8, the retention time of most OSs listed is still less than 1 minute. Authors need think more about it. Otherwise, they should clearly claim that the aim of this work is to improve the measurement of specific compounds (i.e., $C_5H_{11}SO_7^-$).

2. Following Comment 1, there also exist co-elution phenomena for OSs standards by the fact that the retention of time of OSs standards (*m/z* 148-372) is less than 1 minute. Did authors compare the signal (or area in MS) of pure standard alone to the mixing standards to evaluate the effect of co-elution.

3. It is better to give the detailed equations or calculation processes when extrapolate the result of detection limits in instrument (ug/mL) to that in the atmosphere (ng/m$^3$).

4. Line 184-186. It is better to show the standard curves.

5. Figure 1. The specific value for *m/z* $HSO_4^-$ should keep same. *m/z* 96.9 and *m/z* 97.1 can not be assigned to the same fragment ion in high resolution MS.

6. Figure 1 and throughout the manuscript: The *m/z* values and concentration values must report the same correct number of significant figures.

7. "m/z" and "k" should be italic. Line 239: "$SO_4^-$"should be "·$SO_4^-$"; Line 269: "5.24.6.07" should be "5.24, 6.07". Authors should also carefully check and correct other typos and grammar errors that are not listed here.

---

## Author Response (AR1)

**Response to reviews**

We greatly appreciate the time and effort you dedicated to thoroughly reviewing our manuscript. We have carefully considered each point and have made revisions accordingly. We believe that your input has significantly improved the manuscript. Below are our responses to each comment.

**Reviewer: 1**

*1.Authors pointed out that the previous method has the co-elution problem that affecting the quantification of OSs, especially for lower-molecular weight and highly polar OSs. Thus, they employed the method of HILIC using amide stationary phase to measure OSs, finding that this method can successfully separate some isoprene-derived OSs (i.e., $C_4H_7SO_7^-$ and $C_5H_{11}SO_7^-$) from other atmospheric OA components. However, as shown in Tables 7 and 8, the retention time of most OSs listed is still less than 1 minute. Authors need think more about it. Otherwise, they should clearly claim that the aim of this work is to improve the measurement of specific compounds (i.e., $C_5H_{11}SO_7^-$).*

Reply:

The separation of typical organosulfates (OSs) such as $C_5H_{11}SO_7^-$ ($m/z$ 215) and $C_4H_7SO_7^-$ ($m/z$ 199) was notably enhanced using this method, as illustrated in Fig. 1, which compares the separation with the previous reversed-phase column. Specifically, for $C_5H_{11}SO_7^-$ ($m/z$ 215), the separation of six peaks by this method is superior to reversed-phase chromatography, in which these IEPOX-derived OSs isomers co-elute in two peaks (Stone et al., 2012). The resolution of isomers is significant, because $C_5H_{11}SO_7^-$ have generated the greatest OSs signal in prior field studies (Froyd et al., 2010; Lin et al., 2013) and may prove useful in elucidating different OSs formation pathways.

[Figure]

**Figure 1.  Comparison of the effects of separation of *m/z* 199 ($C_4H_7SO_7^-$) and *m/z* 215 ($C_5H_{11}SO_7^-$) using the previous method and this work.**

Due to co-eluting effects, the retention time for *m/z* 139, 153, 155, 167 and 169 under the traditional method was 1.30 min (Stone et al., 2012). However, employing the HILIC method, significant shifts in retention times were observed, Specifically, retention times for *m/z* 139 were 0.83 & 1.58 min, *m/z* 153 were 0.79 & 0.82 min, for *m/z* 155, 167, and 169 were 10.48, 0.69 & 1.00 and 1.46 min respectively. Additionally, Fig. 2 displays chromatograms of isoprene organosulfates (iOSs) with retention times of less than 1 min. While some co-elution persists, their retention times do not precisely overlap. This observation underscores the method's potential for effectively separating lower molecular weight and highly polar OSs.

[Figure]

**Figure 2. Chromatograms of iOSs with retention times less than 1 min.**

*2*. *Following Comment 1, there also exist co-elution phenomena for OSs standards by the fact that the retention of time of OSs standards (m/z 148-372) is less than 1 minute. Did authors compare the signal (or area in MS) of pure standard alone to the mixing standards to evaluate the effect of co-elution.*
Reply:

In this experiment, six OS standards were analyzed. Table 1 compares the retention times and peak areas of pure and mixing standards. The results indicate that the retention times for all standards remained unchanged. Furthermore, there was no co-elution observed between the pure and mixing standards of small molecular weight iOSs, such as $CH_3SO_4^-$ & $C_2H_5SO_4^-$. The peak area ratios of pure to mixing standards were 1.00 and 0.96, respectively. However, co-elution exists for the long-chain alkane OSs ($C_{12}H_{25}SO_4^-$, $C_{16}H_{33}SO_4^-$, $C_{18}H_{37}SO_4^-$), with peak area ratios of 0.57, 0.60, and 0.67, respectively. The mixing standards reduced the signal by almost half, possibly due to a retention time of approximately 0.5 min, falling within the column deadtime.

The ratio of the standards with retention time (*t*R) were 0.8-1 min are close to 1, showing that even

though some of the standards closely elute this doesn't effect the instrument response, suggesting no matrix effect. But the long chain OSs, which elute in the dead volume have a large matrix effect. Meaning that the small amount of retention in this method is much better than the no retention in the reverse phase method. This is also give a evidence to comment 1.

This observation suggests that the analytical effectiveness of this method on iOSs with high polarity surpasses that of long-chain alkane OSs.

**Table 1. Comparison of retention time and peak aera in MS between pure standards and mixing standards.**

| Compounds | [M-H]⁻ | | Standards | $t$R (min) | Peak area | Peak area ratio |
| | $m/z$ | Formula | | | | (Pure/mixing) |
|---|---|---|---|---|---|---|
| Sodium methyl sulfate | 111 | $CH_3SO_4^-$ | pure | 0.92 | 19059629 | 1.00 |
| | | | mixing | 0.92 | 19009710 | |
| Sodium ethyl sulfate | 125 | $C_2H_5SO_4^-$ | pure | 0.81 | 15696871 | 0.96 |
| | | | mixing | 0.81 | 16315513 | |
| Sodium octyl sulfate | 209 | $C_8H_{17}SO_4^-$ | pure | 0.56 | 44588250 | 0.86 |
| | | | mixing | 0.56 | 51744174 | |
| Sodium dodecyl sulfate | 265 | $C_{12}H_{25}SO_4^-$ | pure | 0.52 | 34579898 | 0.57 |
| | | | mixing | 0.52 | 60595452 | |
| Sodium hexadecyl sulfate | 321 | $C_{16}H_{33}SO_4^-$ | pure | 0.51 | 31064839 | 0.60 |
| | | | mixing | 0.51 | 51815669 | |
| Sodium octadecyl sulfate | 349 | $C_{18}H_{37}SO_4^-$ | pure | 0.50 | 36757474 | 0.67 |
| | | | mixing | 0.50 | 55209165 | |

*3*. *It is better to give the detailed equations or calculation processes when extrapolate the result of detection limits in instrument (ug/mL) to that in the atmosphere (ng/m³).*

Reply:

Thanks for your suggestion! We also add these equations in the main text, see new line 233-234.

$$MDLs = IDLs * \frac{V_1}{V_2} \tag{1}$$

$$V_2 = V_0 * \frac{S_1}{S_2} \tag{2}$$

Where IDLs is instrument detection limits, MDLs is method detection limits. The area of a sampling filter (82mm diameter) for OSs analysis ($S_1$) was 52.78 cm², and the total area of a sampling filter ($S_2$)

was 411.84 cm$^2$. The total air volume of 4 h sampling at a flow rate of 1.13 m$^3$ min$^{-1}$ (V$_0$) was 271.2 m$^3$, the solution volume in the vial for LC/MS analysis (V$_1$) was 300 μL, which same as the volume for internal standard added, and the air volume responding to the filter analyzed (V$_2$) was 34.76 m$^3$.

*4. Line 184-186. It is better to show the standard curves.*

Reply:

Thanks for your suggestion! Also see new line 226.

[Figure]

**Figure 3. Correlations between concentration ratios and area ratios of standards to the internal standard, r is the correlation coefficient.**

*5. Figure 1. The specific value for m/z HSO$_4^-$ should keep same. m/z 96.9 and m/z 97.1 can not be assigned to the same fragment ion in high resolution MS.*

Reply:

We revised, see new lines 293 and Fig.4.

*6. Figure 1 and throughout the manuscript: The m/z values and concentration values must report the same correct number of significant figures.*

Reply:

We revised, see new lines 293 and Fig.4, and we also revised that throughout the manuscript.

*7. "m/z" and "k" should be italic. Line 239: "SO$_4^-$"should be "·SO$_4^-$"; Line 269:"5.24.6.07" should be "5.24, 6.07". Authors should also carefully check and correct other typos and grammar errors that are not listed here.*

Reply:

We revised, see new lines 286 and 314, and we also checked the full text, see revision with markup for more details.

**Reviewer: 2**

*1. The full name of the abbreviations should be given at the first time the abbreviations appear. For example, "OSs" in Line 31, "ACN" in Line 114, etc.*

Reply:

We revised, see new lines 28 and 115, and we also checked the full text, see revison with markup for more details.

*2: Some figures and tables in the manuscript have not been referenced in the text, such as Table 3 and Figure 1. All figures and tables should be cited and introduced in the text.*

Reply:

We revised, see new lines 213-216 and 290-292.

*3: Line 269: the dot between "5.24" and "6.07" should be a comma.*

Reply:

We revised, see new line 314.

*4: Line 289: the comma in this line should be a dot.*

Reply:

We revised, see new line 333.

*5: In Table 9, the column "retention time" is sometimes aligned with "daytime" and sometimes positioned between "daytime" and "nighttime". Is this a formatting issue? If not, it should be explained in the manuscript.*

Reply:

We are sorry, it's a formatting issue. We revised it, see new Table 10.

**References**

Froyd, K. D., Murphy, S. M., Murphy, D. M., de Gouw, J. A., Eddingsaas, N. C., and Wennberg, P. O.: Contribution of isoprene-derived organosulfates to free tropospheric aerosol mass, Proceedings of the National Academy of Sciences of the United States of America, 107, 21360-21365, http://doi.org/10.1073/pnas.1012561107, 2010.

Lin, Y. H., Knipping, E. M., Edgerton, E. S., Shaw, S. L., and Surratt, J. D.: Investigating the influences of SO2 and NH3 levels on isoprene-derived secondary organic aerosol formation using conditional sampling approaches, Atmospheric Chemistry and Physics, 13, 8457-8470, http://doi.org/10.5194/acp-13-8457-2013, 2013.

Stone, E. A., Yang, L. M., Yu, L. Y. E., and Rupakheti, M.: Characterization of organosulfates in atmospheric aerosols at Four Asian locations, Atmospheric Environment, 47, 323-329, http://doi.org/10.1016/j.atmosenv.2011.10.058, 2012.